# Virtual Touch Sensor Using a Depth Camera

**DOI:** 10.3390/s19040885

**Published:** 2019-02-20

**Authors:** Dong-seok Lee, Soon-kak Kwon

**Affiliations:** Department of Computer Software Engineering, Dong-eui University, Busan 47340, Korea; ulsan333@gmail.com

**Keywords:** virtual touch sensor, depth camera, depth picture, touch interface

## Abstract

In this paper, a virtual touch sensor using a depth camera is proposed. Touch regions are detected by finding each region that consists of pixels within a certain distance from a touch surface. Touch points are detected by finding a pixel in each touch region whose neighboring pixels are closest to surfaces. A touch path error due to noise in the depth picture is corrected through a filter that introduces a weight parameter in order to respond quickly even with sudden changes. The virtual touch sensor is implemented by using the proposed methods for the touch point detection and the touch path correction. In the virtual touch sensor, a touch surface and pixels of a depth picture can be regarded as a virtual touch panel and virtual touch units, respectively. The virtual touch sensor can be applied to wide fields of touch interfaces. Results of simulations show the implementation of the touch-pen interface using the virtual touch sensor.

## 1. Introduction

Convenient interface methods have been actively researched since the popularization of Internet of Things (IoT) devices. Touch interface is the most successful interface to replace existing interface devices such as keyboard and mouse. Touch interface is easy to learn regardless of gender or age. The interface speed using touch interface is twice as fast as conventional physical interfaces [1].

Currently, capacitive sensing methods [2,3,4,5,6,7] are the most used touch detection methods for touch interface. Capacitive sensing detects touch by measuring capacitances of electrodes. When an object such as a finger touches a panel that applies capacitive sensing, the amount of current that flows through electrodes mounted on the panel is changed. Capacitive sensing has the highest touch detection accuracy and speed among touch sensing methods. However, the price of a device using capacitive sensing increases as a panel size increases, since the touch interface device should be combined with the panel. Instead of the capacitive sensing method, methods using an infrared sensor [8,9,10] and an ultrasonic sensor [11,12,13] can be used to detect touch on a large screen. In the touch detection methods using infrared sensors, infrared emitter sensors attached to the screen emit infrared rays and then infrared detection sensors on the opposite side detect the infrared rays. When touch occurs, a touch object blocks the infrared rays, so the infrared detection sensors cannot detect the infrared rays. Touch detection methods using ultrasonic sensors are similar to the methods using infrared sensors. The touch detection methods using infrared or ultrasonic sensors have a limitation as the shape of the screen has to be flat.

In order to detect touch on a non-flat panel, pictures captured by cameras instead of various sensors can be used. Touch detection methods detect an area of the touch object such as a fingertip in captured pictures for the touch region. The touch point is defined as a pixel that has a highest probability for touching in the touch region. Research by Malik [14] captures pictures from two cameras, then detects a fingertip touching a surface by comparing the pictures. The method used by Sugita [15] detects changes of the color of the fingernail when it touches the surface. However, it is difficult to obtain distance information through color pictures. Therefore, the accuracy of color picture methods is remarkably lower than touch detection methods using sensors. In order to obtain distance information depth pictures, which are captured by a depth camera, can be used. In a depth picture, pixel values are set to distances from a depth camera. Studies on applications of depth pictures have been researched in various fields such as face recognition [16,17,18], simultaneous localization and mapping [19,20], object tracking [21,22,23,24,25], and people tracking [26,27,28]. Interface methods using depth pictures mainly use gesture recognition. Ren [29] proposed Finger–Earth Mover’s Distance, which is a method to measure differences between the hand shapes in depth pictures by applying Earth Mover’s Distance [30], in order to recognize finger gesture. Biswas [31] proposes a gesture recognition method that classifies differences between adjacent frames of a depth video using a support vector machine. Li [32] introduces contour tracing and finger vector calculation to recognize finger gesture. Methods for touch detection using depth pictures have also been studied. Harrison [33] proposes a method where each fingertip is detected by measuring variations of depth values in the vertical or horizontal direction in a depth picture. Touch is detected by measuring differences in depth values between the finger and neighboring areas. Wilson [34] proposes a touch detection method of comparing depth values between each touch object and surfaces. However, these proposed methods are only for touch region detection. Studies for touch point detection in depth pictures are insufficient. It is also necessary to study methods of correcting touch path errors, which are caused by depth picture noise.

In this paper, methods for touch point detection and touch path correction are proposed. Each touch region is detected as a region approaching a certain distance from surfaces, similar to Wilson‘s work [34]. Touch points are detected with pixels closest to surfaces, but other points may also be detected because of depth picture noise. Instead of selecting the closest point to the surfaces, each touch point is detected by finding a pixel in each touch region whose neighboring pixels are closest to surfaces. In touch point tracking, touch path errors occur due to depth picture noise and motion blur. The touch path errors are corrected using a filter similar to the Kalman filter. In the touch path correction using the Kalman filter, there is the disadvantage of a slow response in case of a sudden change of touch point movement. However, the proposed filter solves this problem by introducing a weight parameter that is related to a difference between predicted and measured positions of a touch point. A touch-pen interface is also implemented by the virtual touch sensor.

Surfaces which the depth camera captures and pixels in pictures are set as a virtual touch panel and virtual touch detection units, respectively. Conventional touch interface devices need a screen coupled with sensing devices, while the virtual touch sensor requires only the depth camera as a sensing device. Therefore, the virtual touch sensor is a screen-independent device. 

## 2. Characteristics of Noise in a Depth Picture

Noise in a depth picture occurs due to measurement errors of the depth camera device. This noise causes degrading performance in object feature detection in a depth picture. Figure 1 shows changes in depth values according to frame flow at a certain pixel. The depth values with errors are continuously measured, but an actual depth value is the most frequent. In addition, an average of the depth values is close to the actual depth value.

Shape distortion of a moving object in depth pictures also occurs. Shape distortion is caused by motion blur [35]. Motion blur occurs when the capturing speed of a camera is slower than the speed of the object movement. Figure 2 shows the shape distortion in object detection due to motion blur.

## 3. Virtual Touch Sensor Using Depth Camera

In this paper, a virtual touch sensor using a depth camera is implemented. Surfaces captured by the depth camera are defined as a virtual panel. Pixels of each captured picture are defined as virtual touch units. Figure 3 shows the virtual touch sensor.

### 3.1. Touch Region Detection

In order to detect touch objects using a depth camera, it is necessary to obtain the depth values of a virtual touch panel. Several pictures are used in order to obtain the depth values without noise. The depth values of the virtual panel can be obtained from an average value or a mode value of each pixel in the several pictures. A method using the average values is fast to calculate and simple to implement. However, an incorrect depth value of the virtual touch panel can be obtained if the number of accumulated pictures is not sufficient. A method using the mode values is slow, because a sort for the accumulated depth values in each position is required, however this method can obtain the most accurate depth values of the virtual touch panel. Figure 4 shows methods for obtaining the depth values of the virtual touch panel at a certain pixel using the average and mode values.

The depth values of the virtual touch panel are compared with depth values of each captured depth picture in order to detect touch regions. Depth values of touch objects are different from the depth values of the virtual touch panel, as shown in Figure 5. The touch objects are detected using differences between the depth values of the touch object and the stored depth values for the virtual touch panel as shown in Figure 6b. To consider depth picture noise, regions composed of pixels which satisfy the following equation is set to the touch regions:(1)$$o_{x,y}=\{\begin{array}{ll} 1, & if\;T_{l}<b_{x,y}-d_{x,y}<T_{u}\\ 0, & otherwise \end{array}\;,$$
where *b_x,y_* and *d_x,y_* are depth values of the virtual touch panel and the captured picture in position (*x*, *y*), respectively, and *o_x,y_* is a value in position (*x*, *y*) of a binarization picture, which is a picture for touch region detection. A result for touch region detection is shown in Figure 6c. In touch region detection, regions which are not touch regions may also be detected because of noise. To solve this problem, each detected region whose size is less than *S_min_* should be regarded as part of the virtual touch panel. *S_min_* is determined by considering the resolution of the depth picture. Figure 6d shows the final result of touch region detection by removing noise.

### 3.2. Touch Point Detection

For each touch point a certain pixel where the touch occurs is at the edge of each touch region. To detect touch points, pixels whose distance from the edge of the bounding box of the touch region is within *D_s_* are set to a search region. Pixels satisfying the following equation in the search region are detected:(2)$$b_{x,y}-d_{x,y}<T_{t},$$
where *T_t_* means a threshold for touch point detection. Pixels without accurate touch points may also be detected through Equation (2). In order to detect an accurate touch point, neighboring pixels should also satisfy Equation (2). If pixels satisfy Equation (2) at the position of the colored box in one of the 3 × 3 block patterns in Figure 7, a center pixel of the block is determined as a touch point.

Figure 8 shows depth values in the bounding box of a touch region in Figure 6. Colored boxes are search areas and yellow boxes are pixels satisfying Equation (2) when *D_s_* and *T_t_* are set to 2 and 20, respectively. A circled pixel and neighboring pixels in a vertical direction satisfy Equation (2), so that this pixel is detected as a touch point.

Figure 9 shows the implementation of a virtual touch sensor through the proposed touch point detection. The virtual touch sensor can detect touch even if a virtual touch panel is a curved surface.

### 3.3. Touch Path Correction

In tracking touch points detected by the proposed methods, touch path errors occur from noise and motion blur as shown in Figure 10. 

A proposed filter for correcting touch path errors is similar to the Kalman filter that corrects a measured state through a prediction step, but it introduces a weight parameter in order to respond quickly even with sudden changes. The weight parameter is related to a difference between the predicted and measured positions of a touch point. Figure 11 shows the proposed filter.

The position of a touch point in *n*th frame can be predicted by using a position and a speed in a previous frame as follows:(3)$$\begin{array}{c} p_{p}\left(n\right)=p\left(n-1\right)+v\left(n-1\right)\\ v\left(n\right)\equiv p\left(n\right)+p\left(n-1\right) \end{array},$$
where **p***_p_*(*n*), **p**(*n*), and **v**(*n*) mean a predicted position, a corrected position, and a speed of a touch point in *n*th frame, respectively. A weight parameter *α* is calculated by using the difference between a measured and a predicted position as follows:(4)$$\alpha =\{\begin{array}{ll} \frac{1}{T_{f}}e\left(n\right), & if\;e\left(n\right)\leq T_{f}\\ 1, & otherwise \end{array},$$
where *T_f_* is a filter threshold and *e*(*n*) is a pixel distance between **p***_m_*(*n*) and **p***_p_*(*n*). **p***_p_*(*n*) is completely ignored when *e*(*n*) is greater than *T_f_*. **p**(*n*) is calculated as follows:(5)$$p\left(n\right)=\alpha p_{m}\left(n\right)+\left(1-\alpha \right)p_{p}\left(n\right).$$

Figure 12 shows a touch path correction using the proposed filtering.

### 3.4. Implementation of Touch-Pen Interface

In a touch interface using captured pictures, the coordinate system in a captured picture is different from the screen coordinate system. Therefore, it is necessary to match the coordinates of captured pictures with screen coordinates. Homography transformation can match coordinates between the screen and the picture. Four pairs consisting of matching coordinates in the screen and in the captured picture are required for homography transformation. Each pair is obtained by displaying each dot on the screen and touching it. A touch interface according to the touch position can be implemented after matching the coordinates of the screen and the picture.

A touch-pen interface that draws lines following touch paths is implemented using the virtual touch sensor. Scenarios of the touch-pen interface are as follows: (1) A depth camera is placed in a position where it can capture the whole of the virtual touch panel; (2) four points are displayed sequentially on a surface of the virtual touch panel and a user touches each displayed point; (3) when the user drags the virtual touch panel, a line is drawn along dragged paths. Figure 13 shows a sequence of the touch-pen interface scenarios.

### 3.5. Limitations

The virtual touch sensor has an advantage that the size or surface type of the virtual touch panel does not affect touch detection. However, the virtual touch sensor has limitations as touch detection performance is dependent on specifications of the depth camera and touch detection on a dynamic virtual touch panel is difficult. This limitation is caused by using pictures captured by a camera. A detection interval for touch cannot exceed the frame rate of a depth camera. If the frame rate is 30 Hz, a detection interval cannot be less than about 33 ms. The precision for touch detection depends on the resolution of the depth camera. 

The proposed virtual touch sensor can only detect touch if a virtual touch panel is not changed, because touch detection uses the pre-stored depth values of the virtual touch panel. A step that modifies the pre-stored depth values if a virtual touch panel is changed can be introduced to detect touch on a dynamic virtual touch panel. However, updating changes to the stored depth values after obtaining the depth values of a changed virtual touch panel take a lot of time. In order to detect touch in a dynamic touch panel in real time, a method for the classification of a virtual touch panel and virtual touch objects without previously stored depth values needs to be studied further.

## 4. Simulation Results

In order to measure the performance of the virtual touch sensor, we used Xtion Pro Live, which is manufactured by ASUS from Taiwan, as a depth camera. A resolution of depth pictures is specified as 320 × 240. For the virtual touch panel we used a screen with a width and height of 2.2 m and 1.7 m, respectively. We set the default angle and distance between the camera and the virtual touch panel as 30° and 1.5 m, respectively. 

We measured the accuracy of the proposed touch-pen interface. A fixed position is touched 20 times. Each position detected as a touch point is displayed in the virtual touch panel as a dot. Each touch position error is measured as a distance between the touched and displayed position. *S_min_*, *T_l_*, *T_u_*, *D_s_*, and *T_t_*, which are parameters of the virtual touch sensor, are set as 30, 3, 30, 4, and 10, respectively.

The average touch point detection time using the virtual touch sensor is measured as 20 ms. However, the actual touch detection interval cannot exceed the frame rate of the camera. Therefore, the detection time of the virtual touch sensor is slower than conventional touch detection devices using capacitive sensing. These have a touch point detection time of 10 ms in the case of self-capacitance or 6 ms in the case of mutual-capacitance [7].

In obtaining depth values of the virtual touch panel, touch position errors according to methods and a number of accumulated pictures is shown in Figure 14. Touch position errors when obtaining depth values using average values are large when the number of accumulated pictures is less than about 200. An improvement of touch accuracy doesn‘t appear if the number of accumulated pictures are more than about 300.

Table 1 shows success rates of touch region detection and position errors of touch points according to *T_l_*. Some touch regions are not detected due to depth picture noise when *T_l_* is less than 3. On the other hand, touch position errors are increased as *T_l_* is more than 4 because some parts of the object regions are detected as the virtual touch panel.

Table 2 shows position errors of touch points according to *T_t_*. The position errors increase linearly as *T_t_* increases. The number of detected pixels increases as *T_t_* increases, so it is hard to detect an accurate touch point.

Table 3 shows position errors according to an angle between the virtual touch panel and the depth camera. Touch point detection is most accurate when the angle is set to 30°. It is difficult to detect an accurate touch point when the angle is too small, because an area of the virtual touch panel occupied in a captured picture becomes too small. In contrast, errors in touch region detection increase when the angle is large. The accuracy of touch point detection increases as the distance is closer. 

In order to measure the performance of correcting touch paths by using the proposed touch path correction, the proposed correction method is compared with the Kalman filter and a low-pass filter. The low-pass filter corrects touch paths through the following linear equation:(6)$$p\left(n\right)=\left(p_{m}\left(n\right)+p\left(n-1\right)\right)/2.$$
Equation (6) means that this filter corrects a position as an average between a measured position in a current frame and a corrected position in a previous frame. Figure 15 shows the responses of filters to touch point movements. The low-pass filter removes part of the changes in touch point movement unconditionally. The Kalman filter is more responsive to changes than the low-pass filter, but it still tends to eliminate large changes. In contrast, the proposed filter responds quickly even if large change occurs.

We drew a shape as shown in Figure 16a, consisting of repetitive vertical and horizontal lines by using the touch-pen interface. Then we measured angles *θ*_1_, *θ*_2_ as shown in Figure 16b, between horizontal and vertical axes from a drawn line, respectively. A smaller angle is selected from two angles. A selected angle can be considered as an amount of touch path error. The angle errors converge to 0 as the path correction is accurate.

Average angle errors of the original path, corrected paths by low-pass filter, Kalman filter, and proposed filter are 12.688°, 10.792°, 6.937°, and 6.813°, respectively. Figure 17 shows the results of correcting the touch path.

Figure 18 and Figure 19 show average angle errors according to *T_f_*. A correction effect is more clear as *T_f_* increases. However, more shape distortion occurs at positions where movements of touch points suddenly change as *T_f_* increases as shown in Figure 19.

## 5. Conclusions

In this paper, a virtual touch sensor was implemented by using a depth camera. The virtual touch sensor showed the most accuracy when *T_l_*, *T_u_*, the distance between a virtual touch panel and the depth camera and an angle were set as 3, 30, 1.5 m, and 30°, respectively. A touch-pen interface using the virtual touch sensor was also implemented. The virtual touch sensor is expected to solve the problem of applying touch interfaces to a large display, which is a disadvantage of existing physical touch methods. Conventional touch detection sensors can only use a flat surface as a touch panel, while the virtual touch sensor can also use a curved surface. The virtual touch sensor has the advantage that it is cheaper than conventional physical touch sensors. We expect the virtual touch sensor to be applicable to other fields such as a motion recognition.

## Figures and Tables

**Figure 1 sensors-19-00885-f001:**
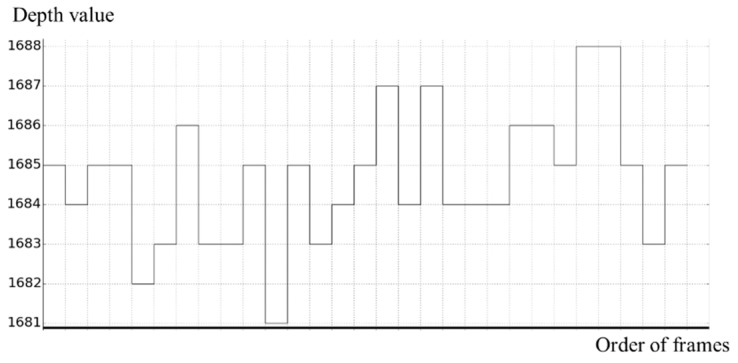
Noises according to frame flow for one depth pixel in depth picture.

**Figure 2 sensors-19-00885-f002:**
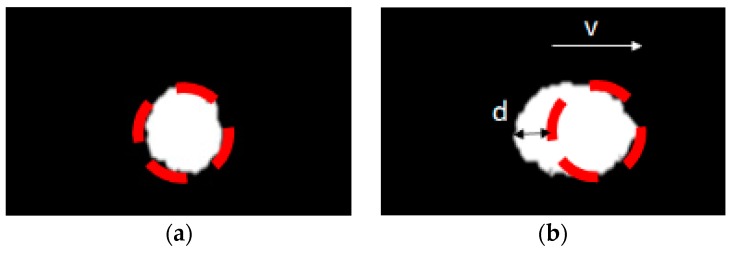
Shape distortion due to motion blur: (**a**) stopping object; (**b**) moving object.

**Figure 3 sensors-19-00885-f003:**
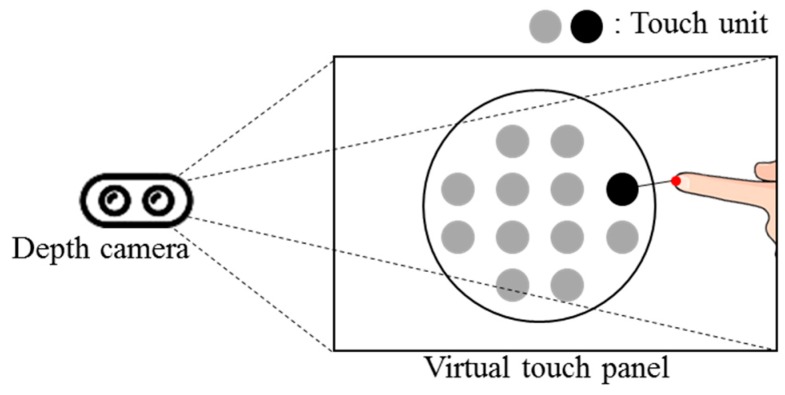
Virtual touch sensor using depth camera.

**Figure 4 sensors-19-00885-f004:**
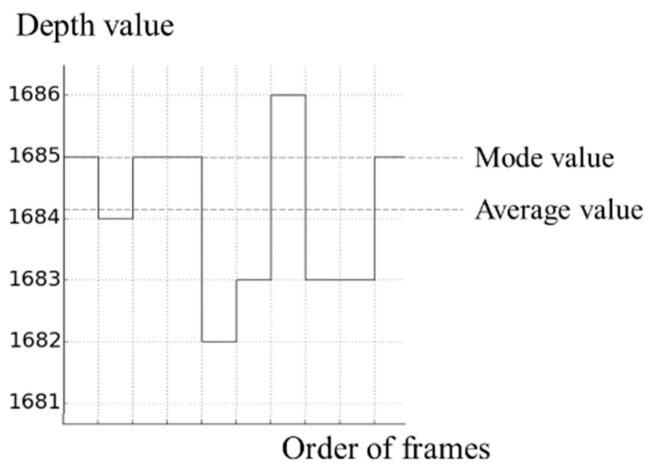
Methods for obtaining depth value of virtual touch panel.

**Figure 5 sensors-19-00885-f005:**
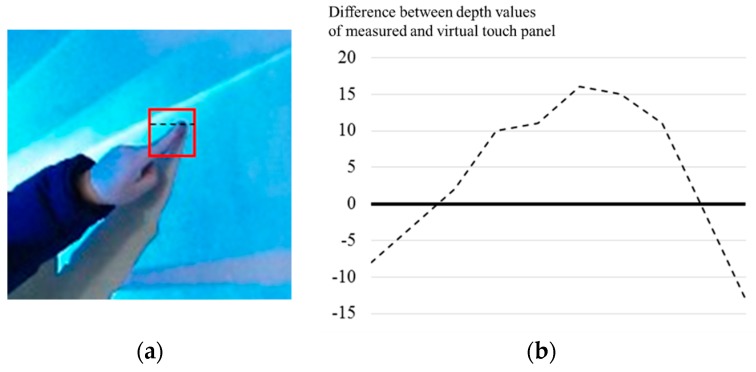
Difference between depth values of measured and virtual touch panel in position of touch: (**a**) original picture; (**b**) difference of depth values from virtual touch panel.

**Figure 6 sensors-19-00885-f006:**
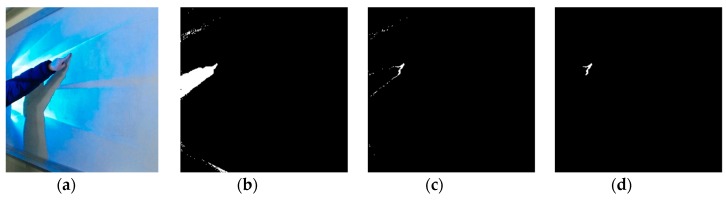
Touch region detection: (**a**) original picture; (**b**) touch object detection; (**c**) touch region detection with noise; (**d**) touch region detection without noise.

**Figure 7 sensors-19-00885-f007:**
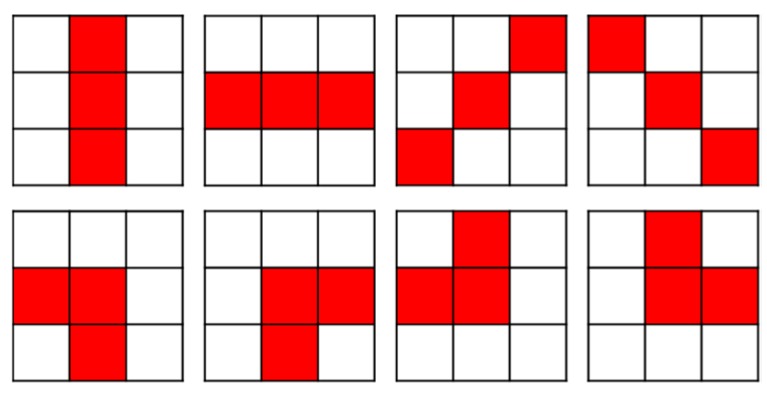
The 3 × 3 block patterns for touch point detection.

**Figure 8 sensors-19-00885-f008:**
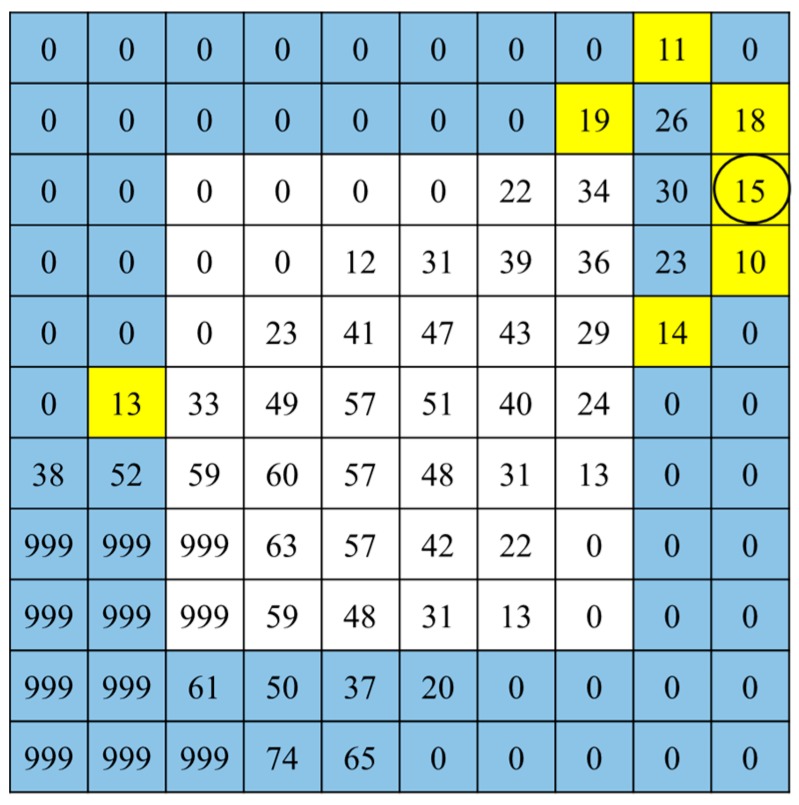
Touch point detection for Figure 6.

**Figure 9 sensors-19-00885-f009:**
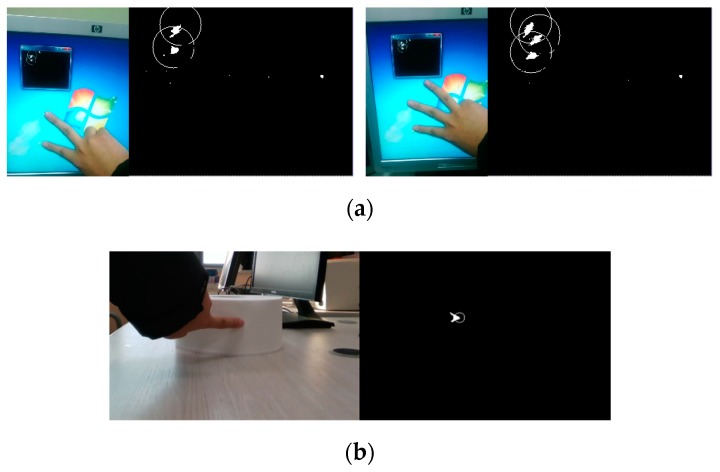
Touch point detection: (**a**) for flat surface; (**b**) for curved surface.

**Figure 10 sensors-19-00885-f010:**
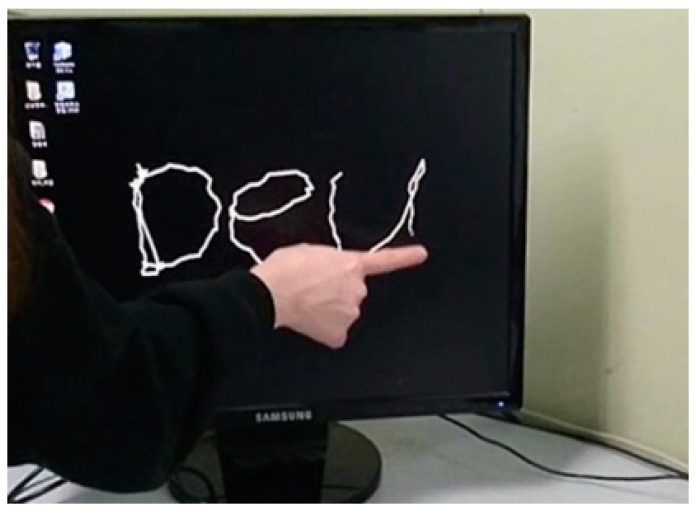
Touch path error.

**Figure 11 sensors-19-00885-f011:**
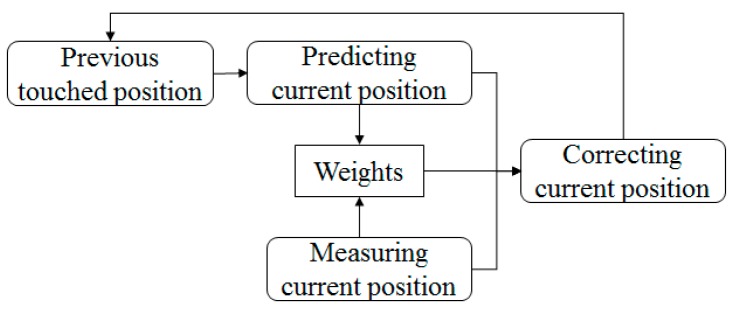
Flowchart of proposed filtering.

**Figure 12 sensors-19-00885-f012:**
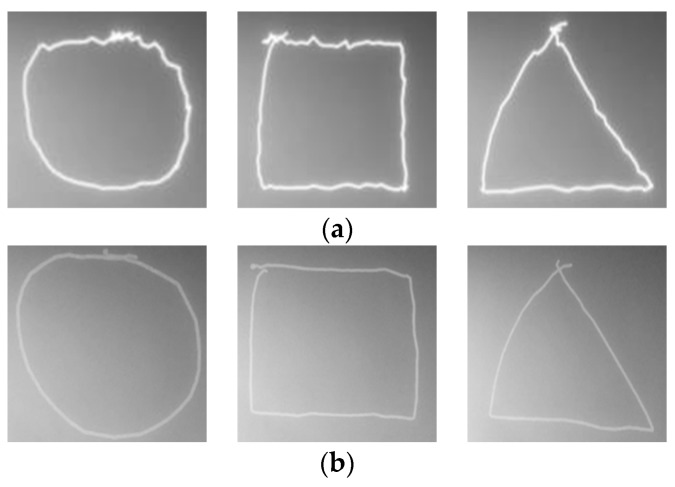
Touch path correction: (**a**) before touch path correction; (**b**) after touch path correction.

**Figure 13 sensors-19-00885-f013:**
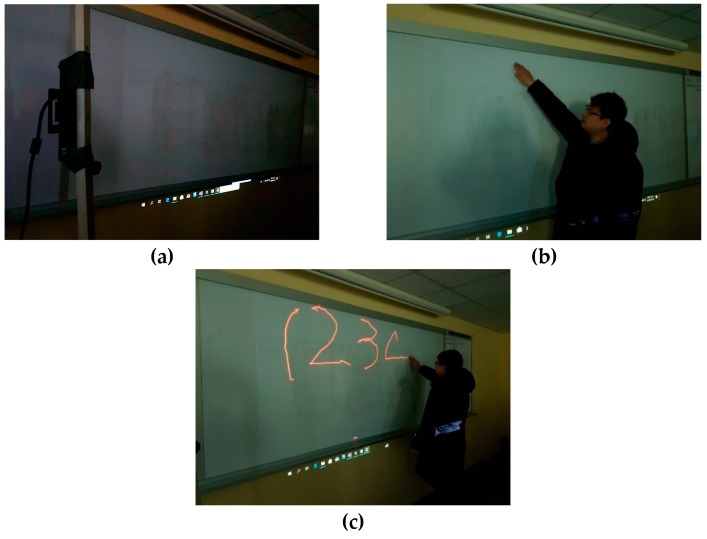
Implementation of touch-pen using virtual touch sensor: (**a**) placing depth camera; (**b**) calibration of virtual touch sensor; (**c**) implementation of touch-pen.

**Figure 14 sensors-19-00885-f014:**
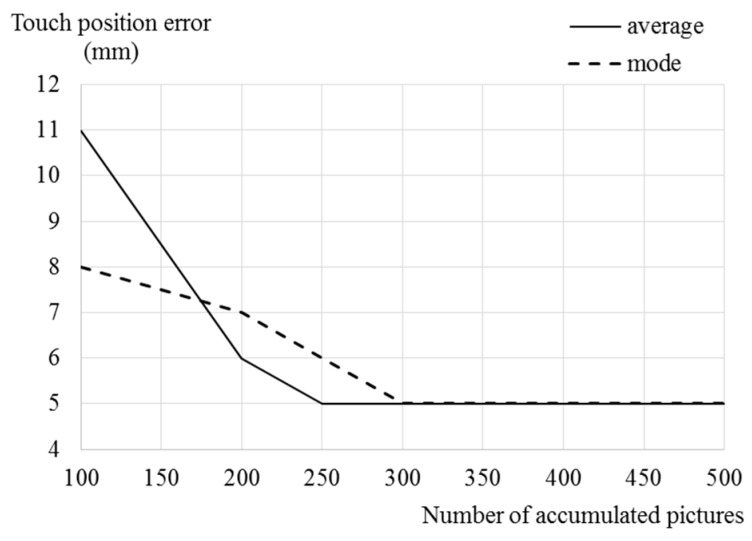
Touch position error according to obtaining methods of depth values of virtual touch panel and the number of accumulated pictures.

**Figure 15 sensors-19-00885-f015:**
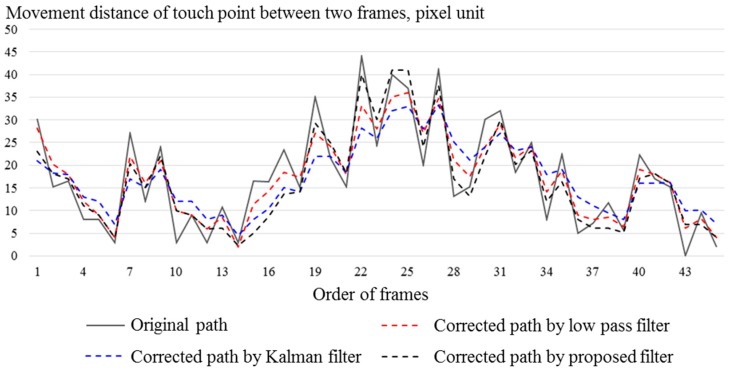
Responses of filters to touch movements.

**Figure 16 sensors-19-00885-f016:**
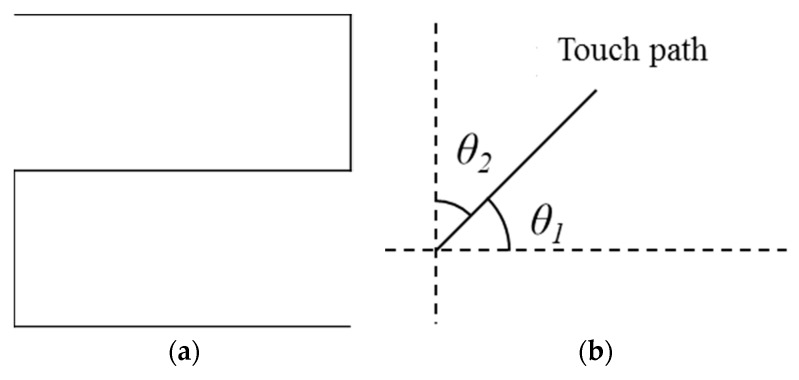
Measuring performance of proposed touch path filtering: (**a**) shape for measuring performance of path correction; (**b**) method for measuring touch path error.

**Figure 17 sensors-19-00885-f017:**
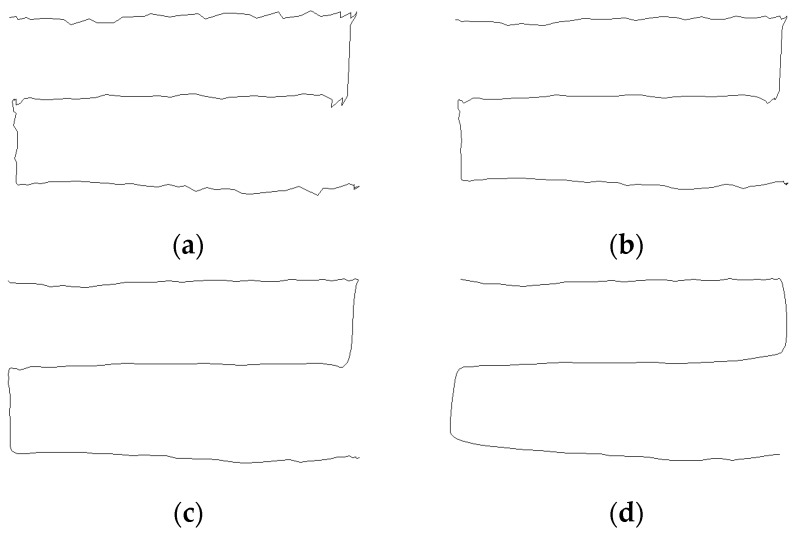
Path correction results according to filtering methods: (**a**) original touch paths; (**b**) correcting by low-pass filter; (**c**) correcting by Kalman filter; (**d**) correcting by proposed method.

**Figure 18 sensors-19-00885-f018:**
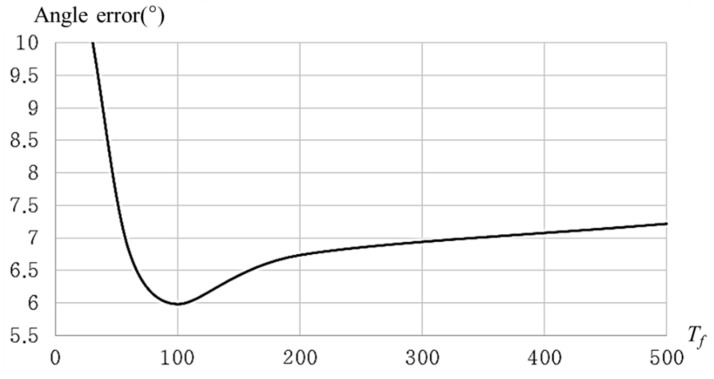
Angle error according to *T_f_*.

**Figure 19 sensors-19-00885-f019:**
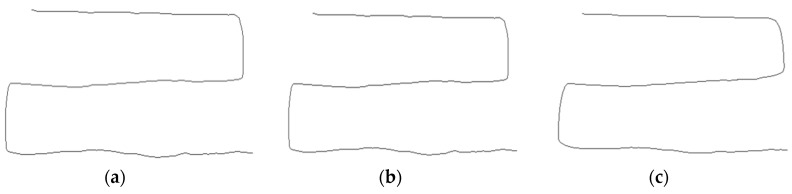
Shape distortion according to *T_f_*: (**a**) *T_f_* = 30; (**b**) *T_f_* = 100; (**c**) *T_f_* = 200.

**Table 1 sensors-19-00885-t001:** Success rate and position errors according to *T_l_* in touch region detection.

*T_l_*	Success Rate of Touch Region Detection (%)	Position Error of Touch Point(mm)
1	0	-
2	75	8
3	100	5
4	100	6
5	100	7

**Table 2 sensors-19-00885-t002:** Position errors according to *T_t_* in touch region detection.

*T_t_*	Position Error of Touch Point (mm)
1	3
2	5
3	7
4	9
5	11

**Table 3 sensors-19-00885-t003:** Position errors according to distance and angle between camera and screen.

Angle(°)	Distance between Camera and Screen (m)				
	1.5	2	2.5	3	3.5
10	16	17	18	18	19
15	12	13	13	14	14
20	8	8	10	12	12
25	5	6	9	10	10
30	5	6	6	7	7
35	6	6	6	7	7
40	7	7	8	8	9
45	8	9	9	10	13
50	9	11	13	15	17
55	10	12	13	16	18

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
