# Peer review of "Virtual Touch Sensor Using a Depth Camera"

_sensors, 2019, doi:10.3390/s19040885_

Round 1
Reviewer 1 Report
This paper presents a simple yet effective method for detecting multi-touch events in large virtual screen areas. The problem is interesting and relevant in the field of touch screens since current capacitive ones provide inaccurate multi-touch detection. Rather than tackling the problem from a hardware/touch screen approach, the paper proposes the use of any area (such as a regular screen or a wall) as touch screen and the use of a depth camera for touch event detection. The paper details all image processing stages for such purpose (depth value, binarization, labeling, and filtering) and presents experimental results mainly on touch detection probability and positioning error.
Overall, the paper presents an interesting topic for the journal. However, I have some remarks:
1) The paper is not easy/fluent to read. There is a significant number of grammatical errors throughout the paper. It needs a deep revision before acceptance.
2) Page 4: It is clear the two ways of obtaining the depth values: average and mode (Figure 4). However, the idea on lines 125-126 and figure 5 are not clear. Here, we have depth value vs x axis, why is depth behavior linear descendent and the touch area dotted line like that? This part needs a more comprehensive explanation
3) It is not clear why Fig. 6 is a false detection (we can appreciate a standard binarization of a 2-finger hand like the one in fig. 9). It would be convenient to include a true detection image besides this false detection one in Fig. 6 and discuss.
4) Figure 7 cannot be well understood without having the original RGB image. Fig. 7 should include: (a) original image, (b) binarization for noisy situations, and (c) removing noise.
Author Response
We express our appreciation for your repeated review. We believe that your detailed comments and suggestions have contributed substantially to improve the presentation of our study, as well as its overall quality and the manuscript. Following, we offer replies to the points the reviewer addressed regarding the original manuscript.
Point 1: The paper is not easy/fluent to read. There is a significant number of grammatical errors throughout the paper. It needs a deep revision before acceptance.
Response 1: We checked and revised whole of the manuscript to correct for ambiguous expressions and grammatical errors including mentioned grammatical errors. The corrected parts are highlighted in revised manuscript.
Point 2: Page 4: It is clear the two ways of obtaining the depth values: average and mode (Figure 4). However, the idea on lines 125-126 and figure 5 are not clear. Here, we have depth value vs x axis, why is depth behaviour linear descendent and the touch area dotted line like that? This part needs a more comprehensive explanation.
Response 2: We change Figure 4.
Point 3: It is not clear why Fig. 6 is a false detection (we can appreciate a standard binarization of a 2-finger hand like the one in fig. 9). It would be convenient to include a true detection image besides this false detection one in Fig. 6 and discuss.
Response 3: We modified the description of binarization(Touch region detection) as follows:
- The obtained depth values of the touch panel value are compared from a captured depth value of a touch object. Depth values of the touch object are different from the depth values of the virtual touch panel as shown in Figure 5. The touch object can be detected using the difference between the depth values of the touch object and the stored depth values of the virtual touch panel as shown in Figure 6 (b). For consider noises of the depth picture, a region where a difference between two pictures is larger than Tl and smaller than Tu is set to a touch region as follows …
We also change Figure 6-7 in order to explain the touch detection process (binarization process) cleanly.
Point 4: Figure 7 cannot be well understood without having the original RGB image. Fig. 7 should include: (a) original image, (b) binarization for noisy situations, and (c) removing noise.
Response 4: We modified Figure 7 as mentioned in Response 3.

Reviewer 2 Report
The paper describes an approach to detect a touching interaction on arbitrary surfaces using a depth camera. The main idea is to apply the binarization technique subtracting between surface depth (virtual touch panel) and the interacting object (touch action). The authors conducted a set of experiment to confirm the improvement of accuracy against the set of parameters used in this method, with an ultimate goal to apply for a large display.
The authors mentioned the advantages of self-capacitance method against the mutual method where both conventional methods either slows or does not support the multi-touch detection. However, it is not clear to me in this context; why the mutual capacitance is slower than the self-capacitance method? In addition to this, I cannot find the scientific evidence, for example, the experiment that compares the proposed method and conventional mutual capacitance method.
The authors discussed over the proposed method, which uses depth pictures to detect a touch interaction and were cited over previous works (e.g., 13-15 in the paper). Although those previous works in the field were using image processing through the color pictures, the method that uses depth image exists and yet, specify the mean of touch using a finger. For example, https://doi.org/10.1145/1936652.1936665, and https://doi.org/10.1145/2047196.2047255. In addition, such touch interaction also seriously investigated so far using fingernail information, e.g., https://doi.org/10.1109/SICE.2008.4654901. The authors should compare the contribution of this study against those previous work.
In the proposed method, the authors described the average value of the depth information to detect the touch interaction. Therefore, such technique obtained the ‘virtual touch panel’ in advance and only apply for the static touch. Do you consider the dynamic screen scenario? Or this is not in the case? In addition, Fig. 7 shows the binary picture and noise removal using S_min. How the S_min is determined? And therefore, how this method applies to the close region, where the finger and the screen are closed to each other? The authors should explicitly describe and discuss over such limitation.
In the touch path filtering, I cannot understand the difference between the proposed method and a well-known Kalman Filter technique? Is there are produced the same result? If not, it is also interesting to compare the result of those methods (suggestion).
In the implementation section, the application information is not enough and misreading. For instance, it is difficult to understand the motivation of ‘touch pen’ and how it is linked to ‘pinch-out’ and ‘pinch-in’ functions. The authors should provide some ‘scenario’ where those interactions can be applied for, and how this is a link to the motivation of the proposed method.
The experiment seems OK, however, the manuscript does not provide any ‘time’ information. How fast of this technique to detect the touch? And how this is better than previous work and conventional methods? Such information is important in order to replicate the study and to discuss the advantage of this method. Is the distance information shown in Table 3 measure between depth sensor to target or a target to screen? The author also mentioned the detect touch with the curved surface in the conclusion. However, this is never explored in the current context. The authors should explicitly explain how to transfer this technique to the curved surface.
** conclusion **
The paper does not provide clear understanding or guidance on how the implemented depth technique could better than the state-of-the-art either within the same domain (work on depth sensor technique) and with mutual capacitance method. Clearly, there is a need to refine the goal and clarify the originality of the proposed method. Such important components should either done with the experiments (time comparison, accuracy comparison, etc.) and explicitly describe both the advantage and limitation of the proposed technique. In summary, the paper is not clear and have a strong contribution in either aspect of 'sensing' and 'interaction' metaphor. Thus, it requires critical additional experiments and important information as described above. I would like to ‘reject’ the current submission as-is and allows the authors to improve the manuscript before submitting again.
Author Response
We express our appreciation for your repeated review. We believe that your detailed comments and suggestions have contributed substantially to improve the presentation of our study, as well as its overall quality and the manuscript. Following, we offer replies to the points the reviewer addressed regarding the original manuscript.
Point 1: The authors mentioned the advantages of self-capacitance method against the mutual method where both conventional methods either slows or does not support the multi-touch detection. However, it is not clear to me in this context; why the mutual capacitance is slower than the self-capacitance method? In addition to this, I cannot find the scientific evidence, for example, the experiment that compares the proposed method and conventional mutual capacitance method.
Response 1: We removed a comparison between self-capacitance and mutual capacitance from the manuscript because this comparison is not relevant to a subject of this paper. Instead, we modified a description for that part as follows:
Currently, capacitive sensing[2-7] are the most mainly used touch detection methods for touch interface. Capacitive sensing detects a touch by measuring capacitances at electrodes. Capacitive sensing detects touch by measuring a change of an amount of current in each electrode mounted on a touch panel. When an object such as a finger approaches a touch panel, an amount of current flowing through each electrode is changed, so that touch can be detected. Capacitive sensing has the highest touch detection accuracy and speed among touch sensing methods. However, a price of the touch interface device using capacitive sensing increases as a screen size increases since the touch interface device should be combined with the screen.
Point 2: The authors discussed over the proposed method, which uses depth pictures to detect a touch interaction and were cited over previous works (e.g., 13-15 in the paper). Although those previous works in the field were using image processing through the color pictures, the method that uses depth image exists and yet, specify the mean of touch using a finger. For example, https://doi.org/10.1145/1936652.1936665, and https://doi.org/10.1145/2047196. 2047255. In addition, such touch interaction also seriously investigated so far using fingernail information, e.g., https://doi.org/10.1109/SICE.2008.4654901. The authors should compare the contribution of this study against those previous work.
Response 2: We added the discussions of the previous works which uses the color or depth pictures as follows:
- A touch detection methods using captured pictures can be used to detect touch on various types of surface shape. S. Malik[14] proposes a method to detect touch by comparing a fingertip positions of pictures captured from two cameras at same time. A work of N. Sugita [15] detects changes in color of fingernail when a finger touches the screen. However, it is difficult to obtain distance information from color pictures, so that the touch detection accuracy is remarkably lower than the touch detection using the physical sensors. Depth pictures can be used instead of the color pictures in order to obtain distance information. Pixel values in the depth picture are set to distance information to depth camera, not color information. Studies on the application of the depth picture have been researched in various fields such as face recognition[16–18], SLAM (Simultaneous Localization and Mapping) [19,20], object tracking [21-25], and people tracking[26-28]. Interface methods using the depth pictures mainly use the gesture recognition. Z. Ren[29] proposed a method for finger gesture recognition by using the Finger-Earth Mover's Distance, which is a method to measure differences between the hand shapes in the depth picture by applying the Earth Mover's Distance[30]. K.K. Biswas[31] proposes a method of gesture recognition, which method classifies difference pictures between adjacent frames of a depth video by support vector machine. Y. Li [32] introduces a method of finger gesture recognition by contour tracing and finger vector calculation. Methods for touch detection using depth pictures have been also studied. C. Harrison[33] proposes a method to detect a finger by measuring variations of depth values in the vertical direction or the horizontal direction in a depth picture, and to detect touch by measuring differences in depth values between the finger and the neighboring areas. A. D. Wilson[32] proposes a touch detection method using a depth camera, which detects touch by comparison between depth values of each touch object and a surface. However, these studies are methods for only the touch region detection, while studies for a touch point detection, which finds a pixel that is actually touched within a touched area, is insufficient. Also, it is necessary to study the method of correcting a touch path error which is caused by noises of the depth picture.
Point 3: In the proposed method, the authors described the average value of the depth information to detect the touch interaction. Therefore, such technique obtained the ‘virtual touch panel’ in advance and only apply for the static touch. Do you consider the dynamic screen scenario? Or this is not in the case? In addition, Fig. 7 shows the binary picture and noise removal using S_min. How the S_min is determined? And therefore, how this method applies to the close region, where the finger and the screen are closed to each other? The authors should explicitly describe and discuss over such limitation.
Response 3: The proposed virtual touch panel is only static screen scenario. We described that at conclusion:
- In this paper, we proposed the virtual touch sensor for only fixed screen. Studies of the virtual touch sensor for a dynamic screen is more needed.
We added the description of S_min as follows:
- … In the touch region detection, a regions which are not touch regions may be also detected because of noises. To solve this problem, detected regions whose size is less than S_min are regarded as the virtual touch panel region. S_min is determined by considering a resolution of the depth picture. Figure 6 (d) shows a final result of the touch region detection with removing noises.
We added a touch point detection section that covers how to find the specific pixel where a touch occurs.
- 3.2. Touch point detection
A touch point, a certain pixel where the touch occurs, is at an edge of each touch region. Pixels whose a distance from the edge of the bounding box of the touch region is within Ds are set to a search region. Pixels satisfying a following equation in the search region are searched: …
where Tt means a threshold for the touch point detection. Pixels where are not accurate touch points may be also detected through Equation (2). In order to detect an accurate touch point, neighboring pixels should also satisfy Equation (2). If pixels satisfy Equation (2) at the position of the colored box in one of a 3×3 block patterns in Figure 7, a center pixel of the block is determined as a touch point. …
Point 4: In the touch path filtering, I cannot understand the difference between the proposed method and a well-known Kalman Filter technique? Is there are produced the same result? If not, it is also interesting to compare the result of those methods (suggestion).
Response 4: We added the comparison between the proposed correction filter and Kalman filter as follows:
- Touch path errors can be corrected by Kalman filter with a state prediction. Kalman filter is a recursive filter that tracks a state of a linear dynamic system containing noises. Kalman filter models an observed system including state noises and observation noises. Kalman filter has a state prediction step that predicts a state using the modeled system and a measurement update step to correct the state. In the the state prediction, a predicted value at a current time t is estimated through a modeled system and a corrected value at t-1. In a measurement update, a measured value at t is corrected by the predicted value and the corrected value at t-1. When Kalman filtering is applied to a touch path correction, it is assumed that a movement of each touch point has a constant velocity. Kalman filter corrects touch paths accurately in general, while it slightly removes a change in each touch point movement when a velocity of a movement changes rapidly.
A touch path error correction filter that gives a weight with consideration of a speed variation is proposed as shown in Figure 11. The proposed path error correction filter gives the weight with consideration of the speed variation. The weight for a measured position is more given than for a predicted position as a change of velocity is larger. The proposed filter has a prediction step and a correction step similar to Kalman filter. …
We also added the simulation results for comparison between the proposed filter and Kalman filter as follows:
- … Kalman filter is more responsive to changes than the low-pass filter, but tend to eliminate them for large changes. In contrast, the proposed filter responds quickly even if large change is occurred. … Average angle errors of an original path, corrected paths by low-pass filter, Kalman filter, and proposed filter are 12.688°, 10.792°, 6.937°, and 6.813°, respectively. Figure 17 shows results of correcting touch path.
Point 5: In the implementation section, the application information is not enough and misreading. For instance, it is difficult to understand the motivation of ‘touch pen’ and how it is linked to ‘pinch-out’ and ‘pinch-in’ functions. The authors should provide some ‘scenario’ where those interactions can be applied for, and how this is a link to the motivation of the proposed method.
Response 5: We added a scenario of a touch-pen interface as follows:
- Scenarios of the touch-pen interface is as follows: (1) A depth camera is placed in a position where it can capture whole of the virtual touch panel; (2) 4 points are displayed sequentially on a surface of the virtual touch panel and a user touches each displayed point; (3) When the user drags the virtual touch panel, a line is drawn along dragged paths. Figure 13 shows a sequence of the touch-pen interface scenarios.
Point 6 The experiment seems OK, however, the manuscript does not provide any ‘time’ information. How fast of this technique to detect the touch? And how this is better than previous work and conventional methods? Such information is important in order to replicate the study and to discuss the advantage of this method. Is the distance information shown in Table 3 measure between depth sensor to target or a target to screen? The author also mentioned the detect touch with the curved surface in the conclusion. However, this is never explored in the current context. The authors should explicitly explain how to transfer this technique to the curved surface.
Response 6 We added the comparison of touch detection time and discussion as follows:
- An average of the touch point detection time using the virtual touch sensor is measured as 20 ms. The detection time is measured by the time taken from depth picture capture to the touch point detection. However, a touch detection interval cannot exceed a frame capture interval of the camera. If a capture interval for a depth camera is 30 Hz, a detection interval is more than 33 ms. Therefore, an actual touch detection time relies on a frame per seconds of the depth camera. The detection time of the virtual touch sensor is slower than conventional touch detection devices using capacitive sensing, which time is 10 ms in case of a self-capacitance or 6 ms in case of a mutual capacitance[7].
We modified Table 3 to clarify the meaning of 'distance'. We added a figure of a touch point detection for curved surface.

Round 2
Reviewer 1 Report
The paper has undoubtedly improved from its previous version. The introduction provides now a more comprehensive discussion on the context and background of the problem. I have appreciated the discussion on the differences between the average and mode depth value, and the improvement of figures 5 and 6. I consider that my remarks have been correctly addressed.
Still, the paper needs to be further revised at language level. I consider that this can be done at the production stage by mdpi. I have no further comments nor objections for its acceptance.
Author Response
Point 1: The paper needs to be further revised at language level.
Response 1: We checked and revised whole of the manuscript to correct for ambiguous expressions and grammatical errors. The corrected parts are highlighted in revised manuscript.

Reviewer 2 Report
Thank you for taking a lot of effort and resubmitting the revised version of the manuscript.
Overall, I am satisfied with the revised version. Most of the questions were answers/added in the manuscript, and thus, significantly improved the readable and contribution.
I only have small questions and suggestion to improve this manuscript if accepted by the editor. (1) In response 3 (revised cover letter), the authors described the limitation of there screen scenario in the conclusion. However, I think it is better to explicitly explain in the dedicated section. For example, add a new section called ‘limitation’ and describes the limitation and suggestion for future improvement/research for the other researchers. Such ‘Studies of the virtual touch sensor for a dynamic screen is more need’ is insufficient answers and need some suggestion about the idea to improve. (2) In response 4, if the Kalman filter and the proposed method is similar, it is better to directly express the different instead of laundry the well-known ‘Kalman filter’ method and conclude as it is similar except the weight parameter. I suggest the authors remove the line number 155 to 164.
In summary, I have happy for the minor revision prior to the accepted this manuscript.
Author Response
We express our appreciation for your repeated review. We believe that your detailed comments and suggestions have contributed substantially to improve the presentation of our study, as well as its overall quality and the manuscript. Following, we offer replies to the points the reviewer addressed regarding the original manuscript.
Point 1: In response 3 (revised cover letter), the authors described the limitation of there screen scenario in the conclusion. However, I think it is better to explicitly explain in the dedicated section. For example, add a new section called ‘limitation’ and describes the limitation and suggestion for future improvement/research for the other researchers. Such ‘Studies of the virtual touch sensor for a dynamic screen is more need’ is insufficient answers and need some suggestion about the idea to improve.
Response 1: We added ‘Limitation’ section and discussed limitations of proposed a virtual touch sensor and suggestion about the idea to improve as follows:
- 3.5. Limitation
The virtual touch sensor has an advantage that a size or a surface type of virtual touch panel does not affect touch detection. However, the virtual touch sensor has limitations that a touch detection performance is dependent on specifications of a depth camera and touch detection on a dynamic virtual touch panel is difficult. A limitation that touch detection performances are dependent on specifications of a depth camera is caused by using pictures captured by a camera. A detection interval for touch cannot exceed a frame rate of a depth camera. If a frame rate of a depth camera is 30 Hz, a detection interval cannot less than about 33 ms. A precision for touch detection depends on a resolution of a depth camera.
The proposed virtual touch sensor can detect touches only if a virtual touch panel is fixed because touch detection uses pre-stored depth values of a virtual touch panel. An update step that modifies pre-stored depth values if a virtual touch panel is changed can be introduced to detect touches on a dynamic virtual touch panel. However, updating changes of the stored depth values spends a lot of time for capturing and obtaining depth values of changed virtual touch panel. In order to detect touch in a dynamic touch panel in real time, a method for a classification of a virtual touch panel and virtual touch objects without previously stored depth values should need to be further studied.
Point 2: In response 4, if the Kalman filter and the proposed method is similar, it is better to directly express the different instead of laundry the well-known ‘Kalman filter’ method and conclude as it is similar except the weight parameter. I suggest the authors remove the line number 155 to 164.
Response 2: We modified the explanation of proposed filter to directly express differences from Kalman filter as follows:
- A proposed filter for correcting touch path errors is similar to Kalman filter that corrects a measured state through a prediction step, but it introduces a weight parameter in order to respond quickly even with sudden changes. The weight parameter is related to a difference between a predicted and a measured positions of a touch point. Figure 11 shows the proposed filter.
